# Improved electrochemical performance of multi-walled carbon nanotube reinforced gelatin biopolymer for transient energy storage applications

**Rabeya Binta Alam, Md. Hasive Ahmad, Muhammad Rakibul Islam** *

Department of Physics, Bangladesh University of Engineering and Technology (BUET), Dhaka, Bangladesh

* rakibul@phy.buet.ac.bd

**Data Availability Statement:** All relevant data are within the paper.

**Funding:** The authors, gratefully acknowledge the financial support from the Bangladesh University Grant Commission under the grant:

## Abstract

Multi-walled carbon nanotube (MWCNT) incorporated biodegradable gelatin nanocomposites (Gel/MWCNT) have been prepared following a facile solution processing method. The Fourier-transform infrared (FTIR) spectroscopy, field emission scanning electronic microscopy (FESEM), and water contact angle (WCA) measurements revealed improved structural properties and surface morphological features of the nanocomposite films due to the incorporation of MWCNT. A four-fold decrease in the DC resistivity was obtained due to the addition of MWCNTs. The specific capacitance of the nanocomposite increased from 0.12 F/g to 12.7 F/g at a current density of 0.3 µA/cm$^2$ due to the incorporation of 0.05 wt.% MWCNT. EIS analysis and the corresponding Nyquist plots demonstrated the contributions of the different electrical components responsible for the improved electrochemical performance were evaluated using an equivalent AC circuit. The incorporation of MWCNTs was found to reduce the charge-transfer resistance from 127 Ω to 75 Ω and increase the double-layer capacitance from 4 nF to 9 nF. The Gel/MWCNT nanocomposite demonstrated improved cyclic stability with a retention of 95% of the initial capacitance even after 5000 charging/discharging cycles. The biodegradability test showed that the nanocomposite degraded completely after 30 hours of immersion in water. This fully biocompatible nature of the nanocomposites with high specific capacitance and low charge transfer resistance may offer a promising route to fabricate a nature-friendly electrode material for energy storage applications.

## Introduction

The worldwide energy demand is increasing abruptly with the population and economic growth, together with the development and advancement of technology. Nowadays fuels are used to meet around 80% of the energy demand. The higher consumption of fossil fuels leads to the creation of a large amount of greenhouse gas that has detrimental effects on the global environment. As a result, relaience on energy from renewable has escalated. However, energy

37.01.0000.073.07.045.21.944. RBA and MRI are also grateful to the Committee for Advanced Studies and Research (CASR), Bangladesh University of Engineering and Technology for providing financial support under the grant DAERS/R-01/CASR-337th/2021. The funders had no role in study design, data collection and analysis, decision to publish, or preparation of the manuscript.

production from renewable sources like solar or wind faces several challenges, including reliance on weather and the inability to store and send power when needed [1,2]. Electrical energy storage is therefore required to obtain uninterrupted and consistent power from renewable energy sources. The fabrication of low-cost and high-efficiency energy storage devices has therefore gained substantial research attention [3–8].

Recently, polymers with good electrical conductivity have found their applications in diverse technical fields because of their cost efficiency, light weight, stiffness, flexibility, etc. More specifically, due to their electrochemical performance, conducting organic polymers have been used in fabricating electrochemical sensors, redox capacitors, catalysis, batteries, etc. [9–12]. In most cases, synthetic polymers are used for the synthesis of functional devices. Polymers obtained from the synthetic routes are non-biodegradable and therefore hazardous to living organisms and the environment. Therefore, it is necessary to synthesize materials from natural and renewable resources to maintain environmental purity and provide a sustainable energy future.

Among various biopolymers, gelatin is considered a popular choice for the environment-friendly transient device applications due to its renewablity, easy-availability, cost-efficiancy, simple processability, and bio-degradability [13,14]. Gelatin is a natural biopolymer extracted from the hydrolysis of collagen stored in the bones and skin of animals [15,16]. It is a water-soluble bio-protein with enriched functional groups, making it preferable to use in transient device applications [17,18]. In this work, we intend to focus on the capacitive performance of gelatin.

The electrochemical performance of gelatin biopolymer can be enhanced by incorporating filler materials into the polymer matrix. The different allotropes of carbon, such as activated carbon [19], carbon nanotubes [20], graphene [21], etc., are known to be chemically stable. They offer large specific surface area for generating electrochemical double layers in supercapacitors and are therefore the popular choice as nanofiller for nanocomposites. Among them, carbon Nanotubes (CNTs), 1D cylinder made from rolled up graphite sheets are one of the most promising nanofiller for polymer nanocomposites [22,23]. CNTs possess different chirality, and based on their formation, they are divided into sub-classes as single-walled and multi-walled carbon nanotubes. Due to their large tensile strength and low production cost, multi-walled carbon nanotubes (MWCNTs) are used more frequently than their single-walled counterparts for synthesizing nanocomposite [24].

Uniform filler dispersion into the polymer matrix is an essential prerequisite to obtain the desired performance from polymer-based nanocomposite. Due to the strong van-der-walls forces among the tube, CNTs agglomerate readily, making it difficult to get a uniform dispersion of nanotubes. Different functional groups such as amino group, hydroxyl group, etc. in the gelatin matrix attach to the unhybridized electron of the MWCNTs and wrap around them, creating a well-dispersed solution [24].

In this work, we used a facile solution casting method to synthesize Gel/MWCNT composite film. Solution casting offers an easy and economical way to synthesize large-scale polymer-based nanocomposite for industrial applications[25,26]. The different properties of the nanocomposite films were investigated by Fourier-transform infrared (FTIR) spectroscopy, field emission scanning electronic microscopy (FESEM), water contact angle (WCA) measurement, and DC resistivity analysis. The capacitive properties were studied by cyclic voltammetry and galvanostatic charging-discharging, followed by electrochemical impedance spectroscopy (EIS) analysisis by an equivalent AC circuit. Bio discarding characteristics of the composite were also tested in an aqueous medium to assess their degradability. This nature-derived, non-hazardous, and bio-degradable materials with enhanced energy storage properties might be used as the electrode material to fabricate environment-friendly energy storage devices.

## Materials and methods

### Materials

MWCNTs with diameter ranges between 20–40 nm, and length of 5–15 μm were obtained from Tokyo Chemical Industry Co., Ltd. Gelatin powder (type B from bovine skin, ~200 Bloom) which was chosen as the polymer host and glycerol (98% pure) was chosen as the plasticizer.

### Synthesis of Gel/MWCNT nanocomposite

At first, 0.1 wt% aqueous solutions of gelatin were made by dissolving an appropriate amount of gelatin into DI water at 50°C. Previously prepared MWCNT solution was then mixed with this gelatin solution at different proportions (0.005, 0.01, 0.02, 0.05 wt %). For higher concentration of MWCNTs the nanotubes can agglomerate and which may results in a decrement of the electrochemical performance and that is why the concentration of nanofillers were kept low. For uniform dispersion of MWCNTs, the Gel/MWCNTs solution was sonicated for 2 hours using a probe sonicator. The glycerol plasticizer was then incorporated into the solution, followed by stirring at 70°C for half an hour using a magnetic stirrer. The solution was then poured onto a glass plate to cool down and kept in the open air to evaporate the excessive water. After 3 days, solid films were formed, and they were peeled out of the dish. The S1 Table in the supplementary section shows the optimized parameters required for the synthesis of nanocomposite.

### Characterization techniques

The chemical composition of the nanocomposites was studied by Fourier transform infrared spectroscopy (FTIR) analysis using a (Shimadzu IRSpirit) spectrophotometer.

The morphology of the surface of the Gel/MWCNT nanocomposite was observed via a field emission scanning electron microscope (JEOL JSM-7600F) at an accelerating voltage of 5 kV. A thin film of gold/palladium was deposited on samples prior to the imaging.

The water contact angle (WCA) was measured at ambient condition and at controlled humidity via a contact angle meter (Apex, India) using the sessile drop technique. At first, a 10 μL drop of water was placed on the surface of the materials and the WCA was evauated via tangential method. For each sample, the WCA was estimated for three different places on the film surface, and the mean value was considered.

Van der Pauw's four-point method was applied to investigate of the DC electrical characteristics of the polymer composite. The resistivity, $\rho$ of the thin films, was calculated using the formula $\rho = 2\pi s(V/I)$ [27]. Where $V$, $I$, and $s$ are the potential difference between the inner probes in volt, the current through the outer pair of probes in ampere, and $s$ is the spacing between the probes in meter, respectively.

The electrochemical performance of the nanocomposites were measured via a CS310 electrochemical working station (Corrtest, China) at room temperature. The measurements was performed via a three-electrode setup consisting of a glassy carbon working electrode, a platinum plate ($1\times1$ cm$^2$) counter electrode, a silver/silver chloride electrode (Ag/AgCl) reference electrode, and 0.1 M KCl was used as an ionic electrolyte solution.

The working electrodes were prepared by depositing the slurry of active material. To prepare the slurry the active materials were mixed with Polyvinyl alcohol (PVAand Dimethyl sulfoxide. Then the rsulting mixtures were then sonicated for few hours followed by deposition on a glassy carbon electrode. Then the working electrodes were dried for several hours at 70 °C. The mass of the active materials were found almost same before and after electrochemical measurements.

## Characterization of Gel/MWCNT nanocomposite

### Structural properties

The chemical bonding properties of the Gel/MWCNT nanocomposites were investigated with FTIR, and the FTIR spectrum for different samples is presented in Fig 1. The prominent absorption peak at 3000–3600 cm$^{-1}$ corresponds to the stretching vibration of the -OH bond. Adsorbed and trapped water molecules from the solution at the processing phase are responsible for this peak [28]. The peak at 2883 cm$^{-1}$ corresponds to the NH- stretching coupled with the hydrogen bonding of Amide-A. And the peaks situated at 2935 cm$^{-1}$ represents the CH- and NH$_4^+$ stretching of Amide-B [29]. The other significant peaks of gelatin are the peptide bonds for Amide I, II, and III. The peak at 1636 cm$^{-1}$ represents the C = O stretching and NH bending vibrations of Amide I. The peak at 1549 cm$^{-1}$ originates from the CN stretching and CO bending vibrations of Amide II. The peak for Amide I at 1110 cm$^{-1}$ generates from the fluctuation of CH$_2$ [30]. The peak intensity was found to be increased due to the incorporation of CNTs. The peak intensity increment and presence of active hydroxyl bonds indicate enhanced interactive sights between the polymer and filler [17].

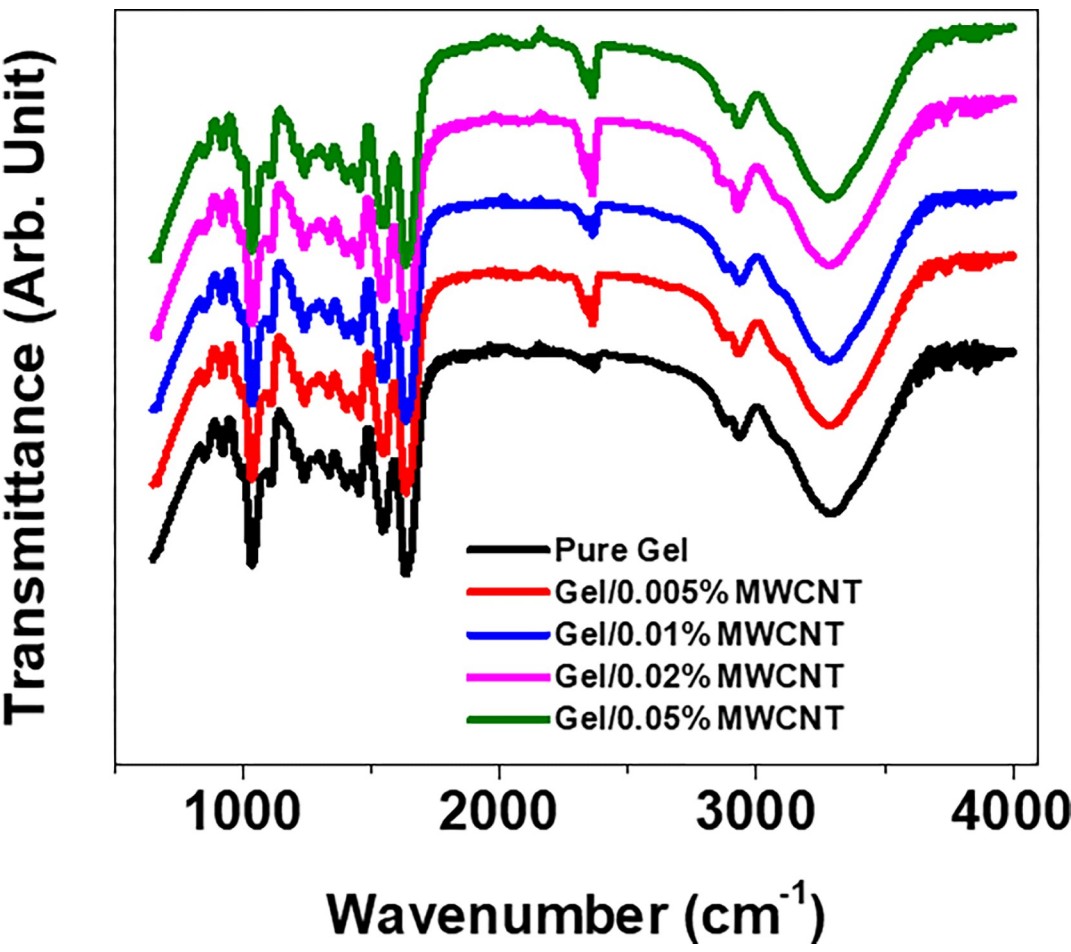

**Fig 1. FTIR spectra of Gel/MWCNT nanocomposites for various amounts of MWCNT filler content.**

## Surface morphology

The FESEM images of the Gel/MWCNT composite for different concentrations of MWCNTs are presented in Fig 2. Pure gelatin film (Fig 2A) is flat, and no trace of any filler material is found there. From the inspection of the FESEM images of the Gel/MWCNT films, displayed in Fig 2B–2F, it is evident that the morphology has changed noticeably compared to the pure Gel film. With the increase of the concentration of MWCNT the roughness of the surface got increased and lots of fibrous pattern are observable on the surface. The Gel matrix might wrapped the MWCNT and create fibrous pattern. These images of the composite materials depict well-porous surfaces with traces of filler materials. Porous surfaces mean a bigger area, which offers better and intimate contact with the electrolyte [31,32]. The observed change implies that the CNTs have been embedded in gelatin with adequate interconnections [33]. Also, the roughness of the nanocomposite enhances due to the incorporation MWCNTssuggesting increasing surface area.

## Surface wettability

The effect of MWCNTs on the surface wettability of the gelatin nanocomposites was evaluated by using the contact angle meter. Fig 3 shows the effect of MWCNT concentration on the water contact angle (WCA) of the nanocomposites. The WCA for the Gel was found to be 67˚ suggesting the hydrophilic nature of the polymer matrix. The WCA of the Gel/MWCNT samples was found to be increase as the amount of MWCNT increases in the polymer. WCA of 120.25˚ were observed for the nanocomposite with 0.05 wt% of MWCNTs suggesting that surface wettability changes from hydrophilic to hydrophobic because of the incorporation of MWCNTs. The non-polar carbon nanotubes are highly hydrophobic, and as a result, adding a minute amount of nanotube changes the nature of the wettability of the polymer matrix [34,35]. From the FESEM images, it is evident that the incorporation of MWCNTs enhances

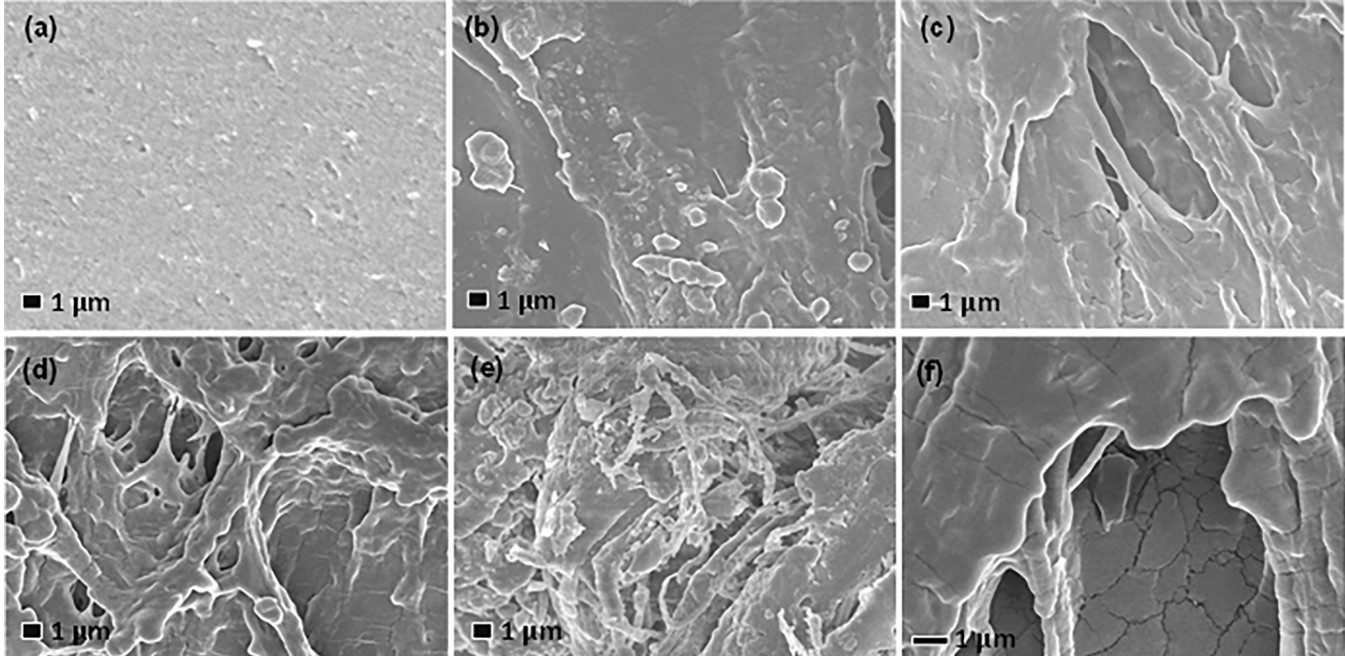

**Fig 2.** FESEM images of the Gel/MWCNT nanocomposites with (a) 0 wt%, (b) 0.005 wt%, (c) 0.01 wt%, (d) 0.02 wt% and (e) 0.05 wt% of MWCNT concentrations at ×5000 magnification (a-e). Fig (f) shows the FESEM images for the Gel/0.05 wt% MWCNT nanocomposite.

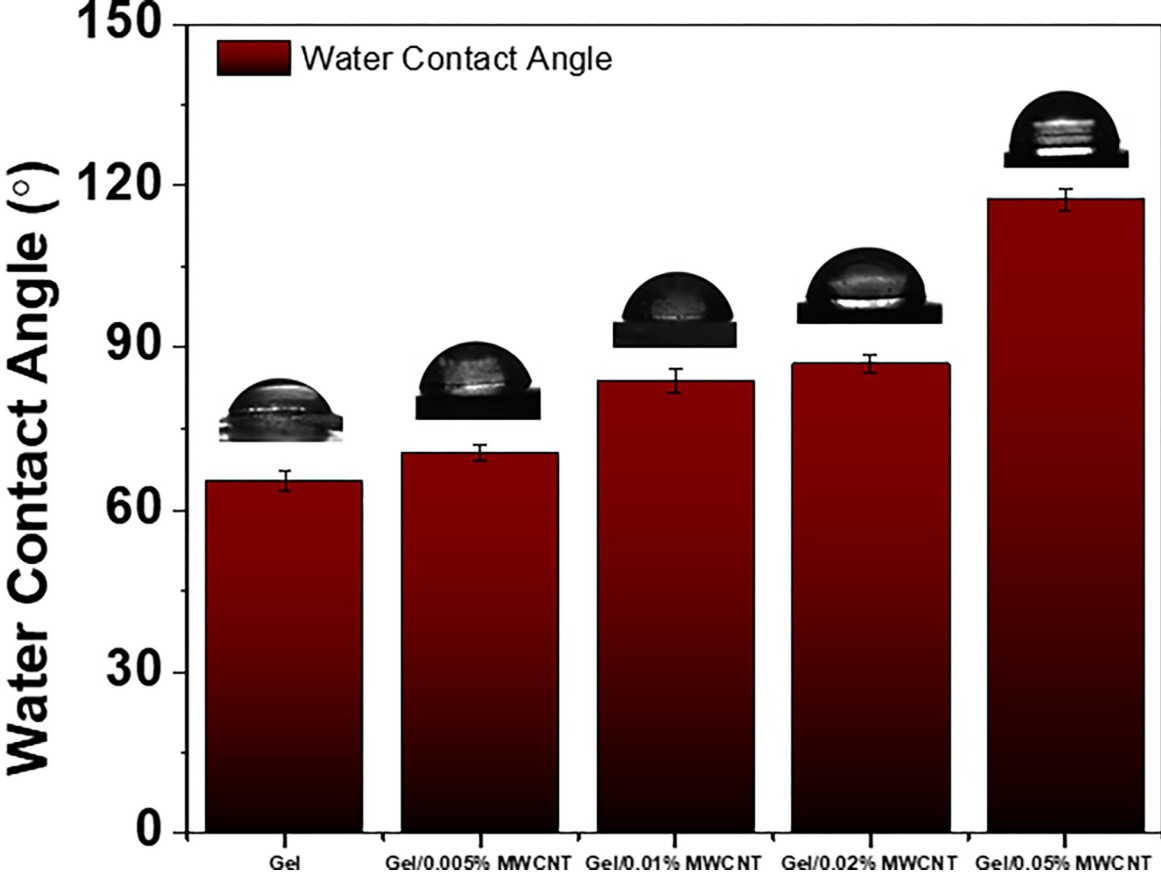

**Fig 3. The change in water contact angle of Gel/MWCNT nanocomposites for different concentration of MWCNTs.**

the roughness of the surface. This increased degree of irregularities may results in the changes in surface wettability of the composite nanomaterials [34,36,37].

## Electrical properties

Fig 4 demonstrates the effect of the concentration of MWCNT nanofiller on the DC resistivity of the gel matrix. Gel is an insulator [36,38] and represents an electrical resistivity of 57 MΩ-cm. Addition of only 0.005 wt% MWCNT into the gel matrix reduces the resistivity to 12 kΩ-cm. The resistivity reduces to 2.5 kΩ-cm for the sample with 0.05 wt% MWCNTs. The MWCNTs nanofillers create conducting channels inside the insulating polymer domain for the charge carriers to pass through. Furthermore, the $sp^2$ hybridized atomic arrangement of the MWCNTs is also responsible for improving the electrical conductivity [22,39]. Besides, the unhybridized electron, under applied voltage, can travel among the tube inside the polymer, resulting in a reduction in resistivity. Processes like hopping or jumping are also responsible for the conduction of electricity throughout the composite via ionic conduction or tunneling [40,41].

## Electrochemical properties

Fig 5A-5E shows the Cyclic voltammetry (CV) curves for the nanocomposites at different scan rates. The specific capacitance $C_S$ (F/g) for the nanocomposites can be obtained from the area

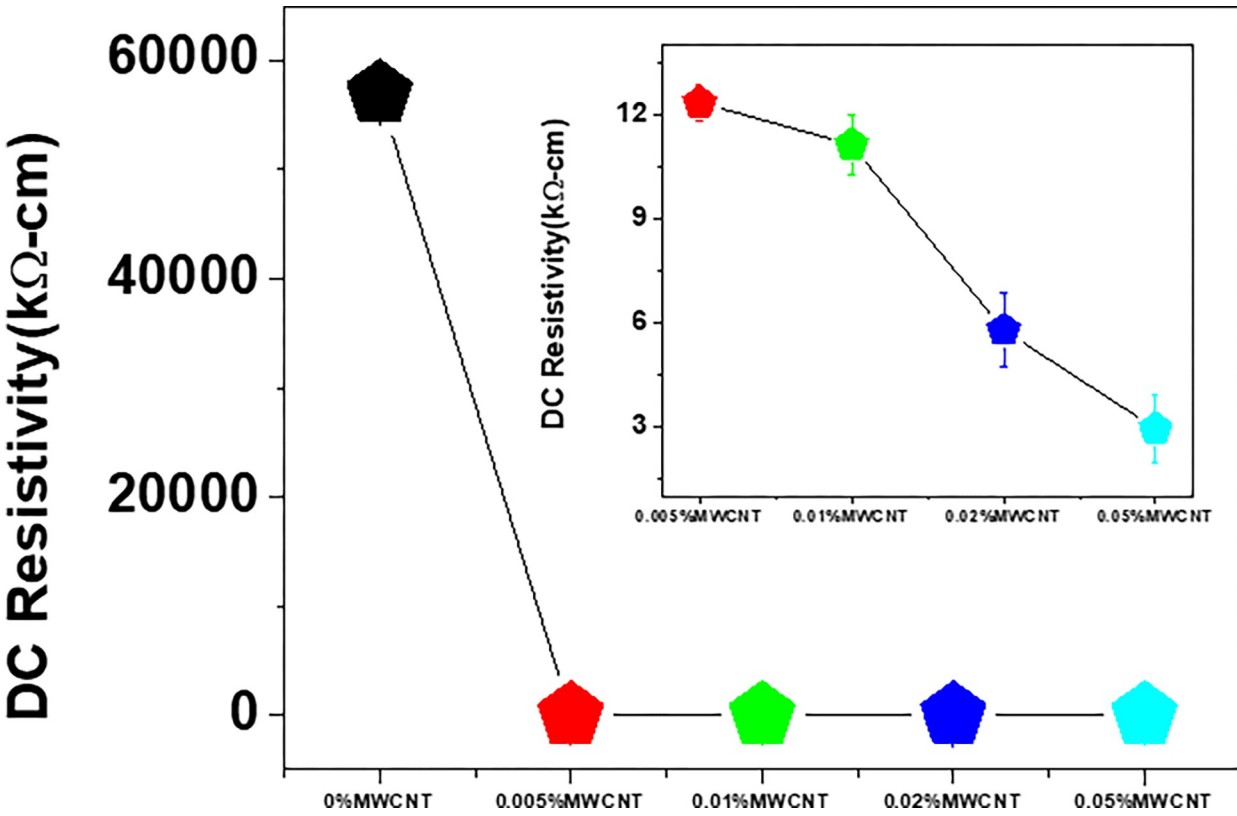

**Fig 4. Variation of the DC resistivity of Gel/MWCNT nanocomposites as a function of MWCNT content.**

under the CV curves following the relation $Cs = \frac{1}{v.m.(V2-V1)} \int_{V1}^{V2} i(V)dV$ [42]. Here, $v$ is the voltage scan rate, $m$ is the amount of electrode materials, and $(V_2-V_1)$ is the applied potential window. The area of the CV curves was found to be increased with the voltage scan rate. At a low scan rate, the electrolyte ions get enough time to diffuse into the intermolecular regions of MWCNTs [42–44]. The CV curves for all the different samples at a particular scan rate (5 mV/s) is presented in Fig 5F. From the figure, it is evident that the area of the CV curves increased with the increase of the MWCNTs concentration in the nanocomposites. This suggests that the composite nanomaterials' capacitance increases with the concentration of MWCNT nanofillers. For samples with higher filler concentrations, a trace of a small bump is observed at around 2.5 V. Such redox curves at the background of CV curves demonstrate the presence of pseudo-capacitance in the Gel/MWCNT sample [45,46]. Thus the improvement of the value $C_s$ for the Gel/MWCNT sample might be due to the formation of a capacitive layer around the nanotube surface together with the presence of pseudo-capacitance in the nanocomposite [47].

Fig 6A–6E represents the Galvanostatic charging and discharging (GCD) curves of the Gel/MWCNT nanocomposites at different current densities. Fig 6F illustrates the variation of the GCD curves due to the presence of nanotube concentration in the nanocomposite. This figure shows that the discharging time increases with the nonofiller concentration. The value of $C_S$ was calculated from the GCD curve using the formula $C_S = \frac{I_{disch}}{m} \frac{dt}{dV}$ [42,45]. Where, $I_{disch}$ and $dV/dt$ represent the current and slope of the GCD curve during discharging. Fig 7 shows how the specific capacitance varies with the current density for the different samples. At lower current densities, ions get enough time to accumulate themselves properly on the MWCNT wall, thus showing higher specific capacitance [48,49]. At higher current densities, the ions cannot

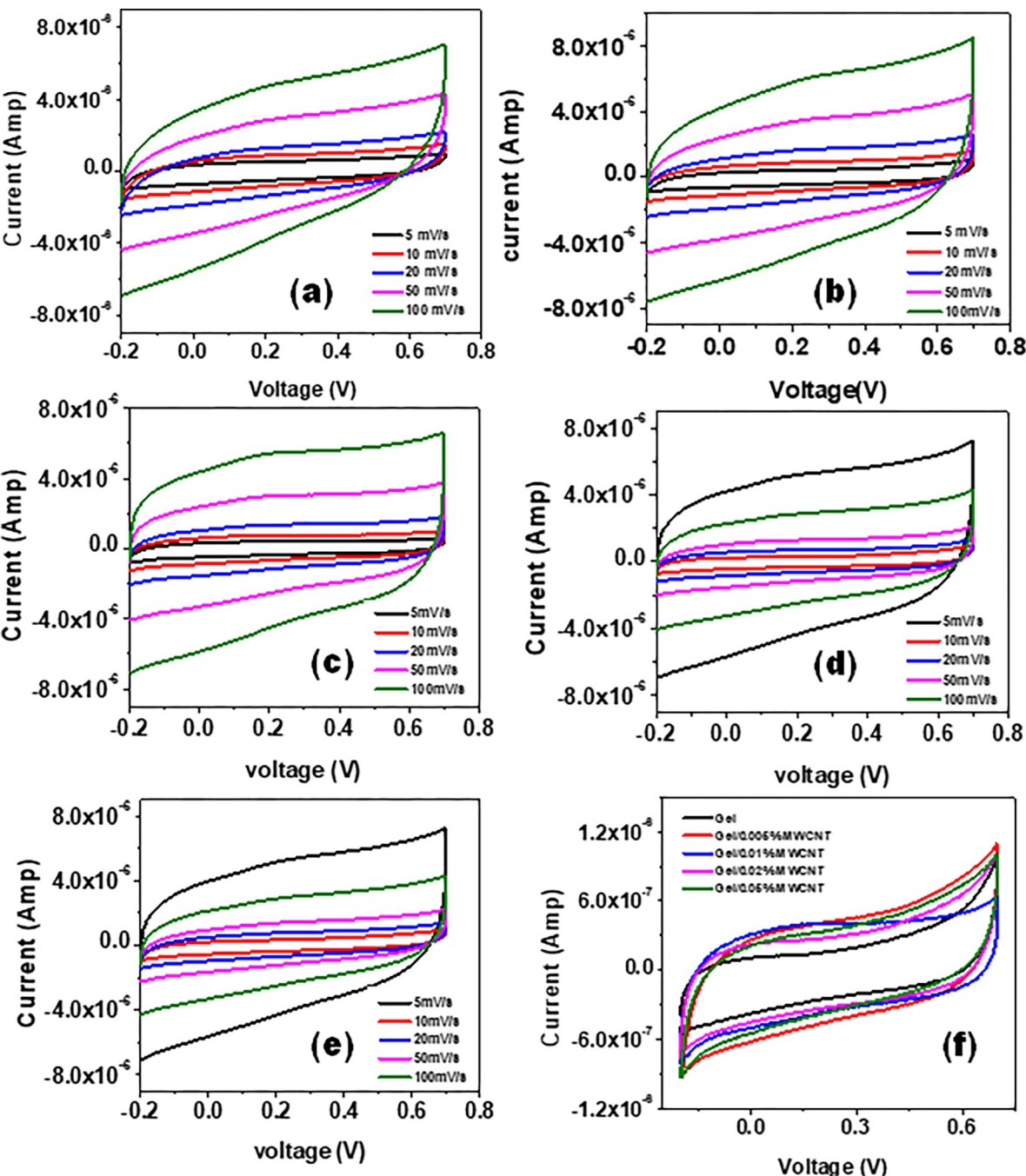

**Fig 5.** Cyclic voltammetry of (a) Gel, (b) Gel/0.005 wt% MWCNT, (c) Gel/0.01 wt% MWCNT, (d) Gel/0.02 wt% MWCNT, (e) Gel/0.05 wt% MWCNT at different voltage scan rates. (f) Comparison of cyclic voltammetry of all the samples at 5 mV/s scan rate.

adhere to the MWCNT surface, thereby reducing the capacitance [48]. From ig 7, it is evident that the value of specific capacitance increases with the concentration of MWCNT in the polymer matrix. For the sample with 0.05 wt% MWCNT, specific capacitances of 12 F/g was obtained at current densities of 0.3 μA/cm$^2$. Increase of two order of magnitude in the specific capacitances was obtained just due to the addition of 0.05 wt.% of MWCNTs into the Gel matrix. Recently, Rabeya et. al. [37] synthesized SWCNT/Gel nanocomposite and showed the

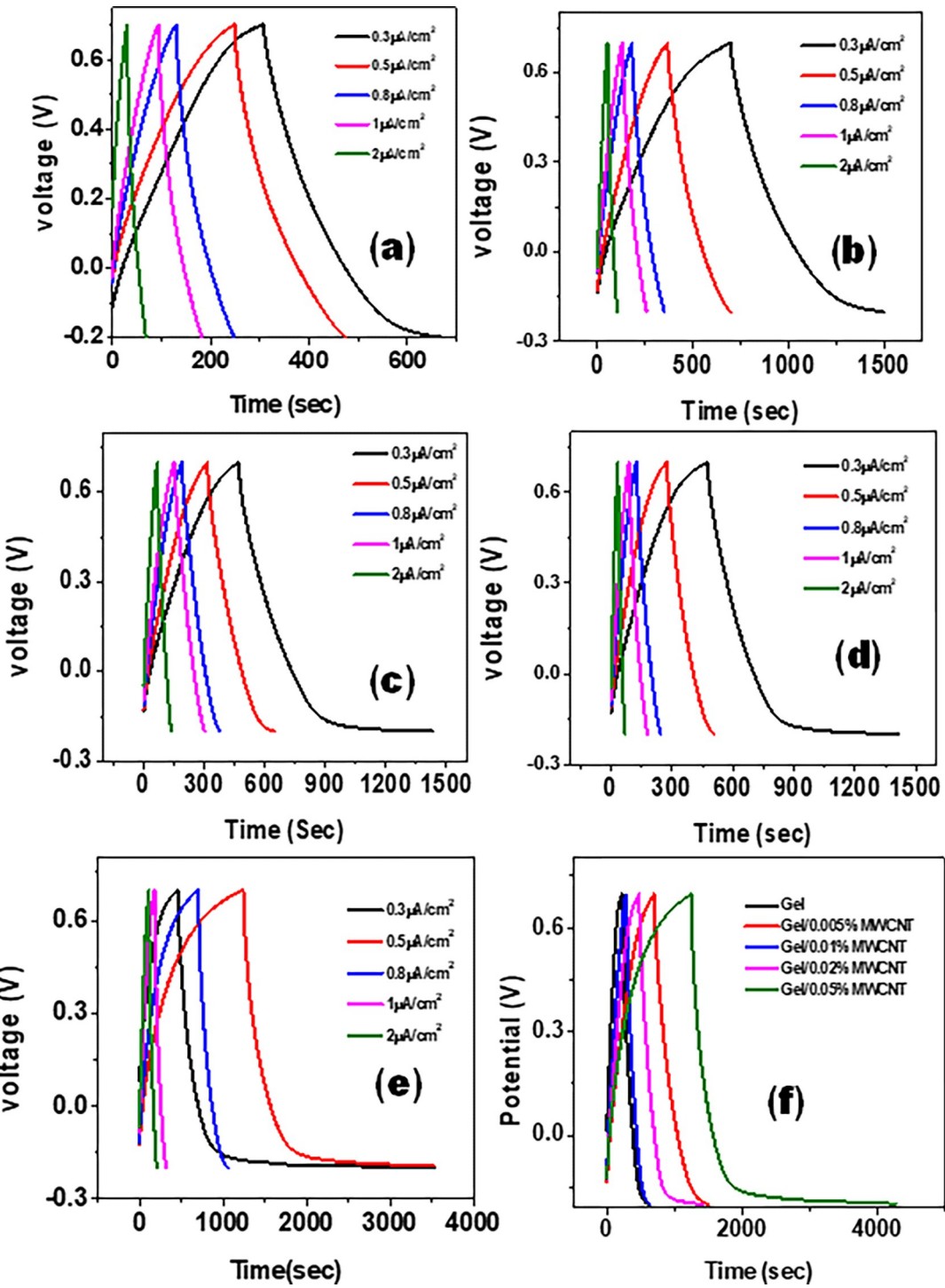

**Fig 6.** Galvanostatic charging-discharging of (a) Gel, (b) Gel/0.005 wt% MWCNT, (c) Gel/0.01 wt% MWCNT, (d) Gel/0.02 wt % MWCNT, (e) Gel/0.05 wt% MWCNT at different current densities. (f) Comparison of GCD curves of all the samples at a current density of 0.3 μA/cm$^2$.

incorporation of SWCNT increase the specific capacitance from 124 mF/g to 467 mF/g. The MWCNTs can therefore be considered as the suitable choice as the nanofiller over SWCNTs for energy storage applications. The MWCNTs form a conductive route throughout the

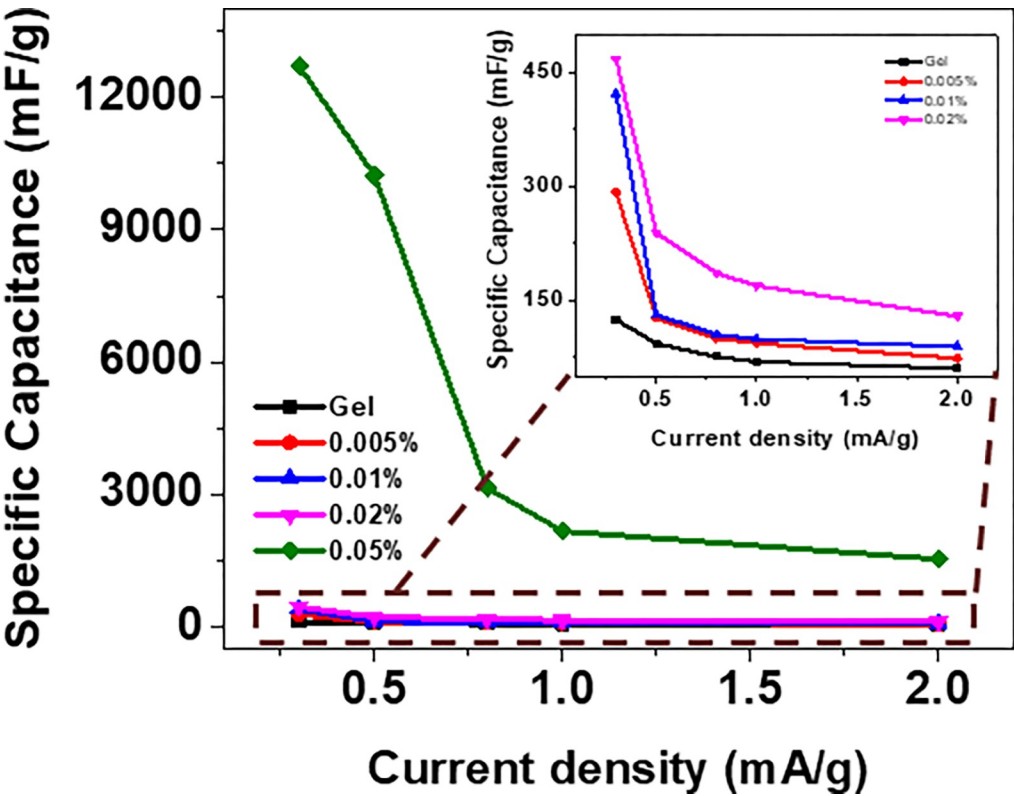

**Fig 7. Calculated specific capacitance from galvanostatic charging-discharging of Gel/MWCNT samples at different current densities.**

polymer matrix, and therby offer enhanced ion transportation [50]. There might be some contribution of pseudocapacitance due to ion absorption and transmission in the polymer chains, which gives the distorted triangular shape of the GCD curves [51].

**Electrochemical impedance spectroscopy.** The electrochemical impedance spectroscopy (EIS) was performed to analyze the electrochemical behavior of charge and ion transportation between the electrolyte and the surface of the material. The EIS data was taken in the frequency range between $10^{-1}$ Hz to $10^6$ Hz and at an AC voltage of 10 mV and is shown in Fig 8. To estimate the ceffect of various electrical components in the Nyquist plot, a simulation was conducted by a network of circuit elements (shown in the inset image of the figure). The solid lines in the figure represent the best-fitted curves, and the estimated values of electrical components are shown in Table 1. The equivalent AC circuit is comprised of an equivalent series resistance (ESR), $R_s$, which corresponds to the resistance of the electrode material, the electrolyte, and contact resistance of the electrode and current collector [21,25]. The series resistance reduces from 8.40 Ω to 3.55 Ω due to the addition of MWCNTs into gelatin. Because of the presence of MWCNTs, the current collectors can quickly move through the polymer matrix and thereby reducing the series resistance. The current collectors can quickly move through the MWCNTs and thereby offer a smaller value of $R_s$ [52]. The charge transfer resistance ($R_{ct}$) can be obtained from the radius of the semicircle in the high-frequency region [53]. The charge transfer resistance also reduces due to the presence of MWCNTs in the polymer matrix. The highly conducting MWCNTs help reduce the value of $R_{ct}$ [53]. The constant phase element (CPE) is attributed to the inhomogeneity and porosity of the superficial electrode [54], and the double layer capacitance ($C_{dl}$) corresponds to the electrical double layer due to the

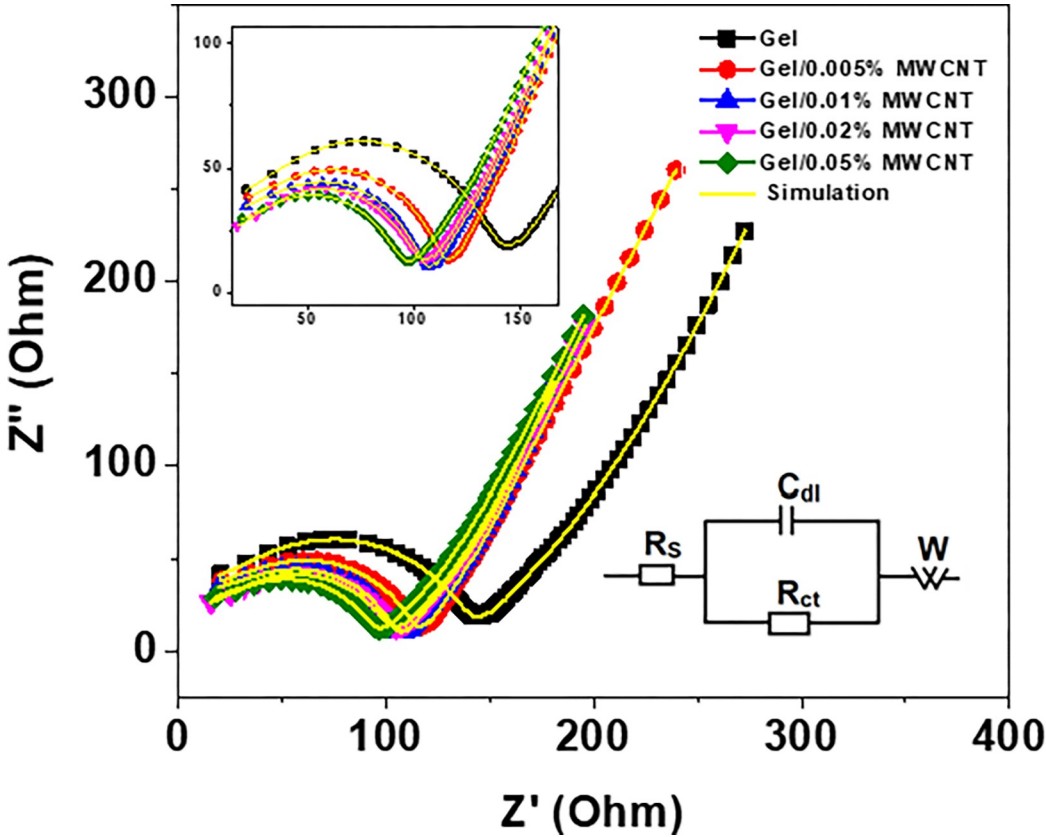

**Fig 8. The complex impedance spectra of the nanocomposites with the fitting curves.** The inset image shows the higher frequency region, and the equivalent circuit for the best-fitted curves.

accumulation of ions at the boundary between the electrolyte and electrode [55,56]. The value of the double layer capacitance was also found to be increased from 4.57 nF to 8.47 nF when 0.05 wt% of MWCNTs were added to the polymer. The MWCNTs filler inside the polymer matrix may create a charge capacitive region, thereby enhance the $C_{dl}$ [20,57]. The Warburg impedance element (W) occurred due to the diffusion of ions from the electrolyte to the surface of electrode material and can be obtained from the slope of the Nyquist plot representing the Warburg impedance element (W) at the mid-frequency regime [58]. The composite material shows a capacitive transformation for incorporating MWCNT in the Gel polymer.

**Cyclic stability.** Cyclic stability is another well-known technique for analyzing the performance of the electrode material. To study the cyclic stability of the Gel/MWCNT nanocomposite, capacitance retention of the Gel/0.05 wt% MWCNT nanocomposite was tested and is

**Table 1. The different circuit components of the equivalent circuit used in simulation of the Gel/MWCNT nanocomposites.**

| MWCNT concentration (wt%) | Series resistance $R_s$ (Ohm) | Double layer capacitance $C_{dl}$ (nF) | CPE exponent $n$ | Charge-transfer resistance $R_{ct}$ (Ohm) |
|---|---|---|---|---|
| 0 | 8.404 | 4 | 0.99 | 126.9 |
| 0.005 | 5.877 | 5 | 0.97 | 97.83 |
| 0.01 | 5.047 | 6 | 0.95 | 91.50 |
| 0.02 | 4.534 | 7 | 0.99 | 81.89 |
| 0.05 | 3.55 | 9 | 0.98 | 75.75 |

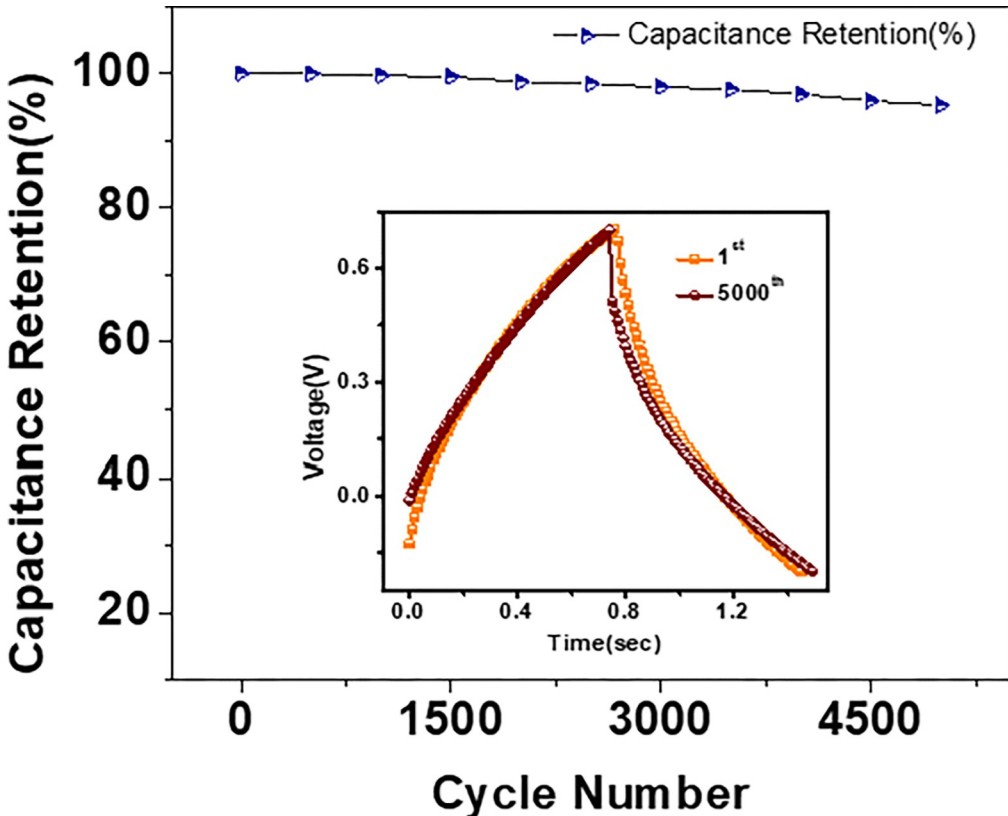

**Fig 9. The capacitive retention for 5000 cycles of GCD for the Gel/0.05% MWCNT nanocomposite.** The inset shows the GCD curves for the 1st and the 5000th cycles.

presented in Fig 9. The sample reveals a large retention of 95% of the maximum capacitance even after completion of 5000 charging/discharging cycles. The inset of the figure demonstrates the GCD curves for the 1st and 5000th cycle and the shape of the curves remains almost unchanged. Such outstanding stability over a large number of cycles ensure that the nanocomposites can be used as a potential candidate as electrode material for energy storage applications [3,59].

## Biodegradability analysis

To study the biodegradability of the Gel/MWCNT nanocomposite, a piece of $30 \times 10 \times 1$ mm$^3$ film from the Gel/0.05% MWCNT sample was kept in simple tap water at room temperature and was monitored. Fig 10A–10H showed that a total degradation of the film occurs within 30 hours of emerging the film into the water. Because of the presence of different hydrophilic groups in the gelatin matrix, they can dissolve and swell in water [36,60]. The nanocomposite film was at first soaked and swollen and then started to break under applying a little stirring. From this observation, it is understandable that the Gel/MWCNT nanocomposite can be discarded in an open environment after its operation without causing any harm to nature.

## Conclusions

We prepared a bio-disposable gelatin-based MWCNT nanocomposite following an easy solution casting procedure. The FTIR and FESEM studies reveal the purity and well dispersion of

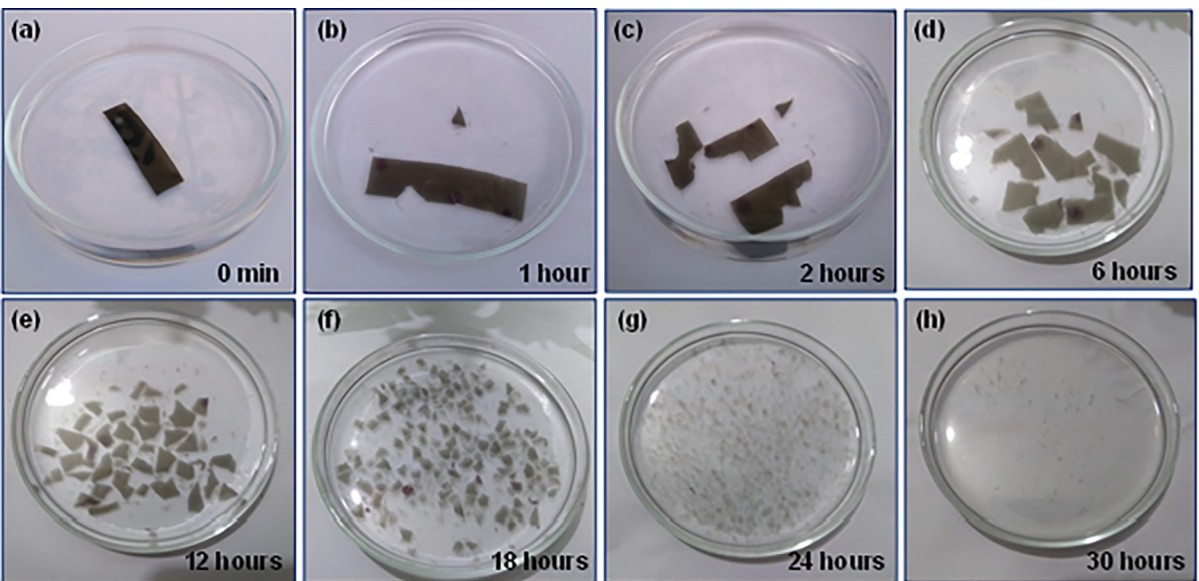

**Fig 10. Biodegradability analysis of Gel/MWCNT nanocomposites in water as a function of time.**

filler material in the nanocomposite. The significant decrease in the DC resistivity shows the improvement of the polymer from its inherent insulator-type behavior due to the addition of MWCNTs. These nanocomposites provide improved electrochemical performance as the specific capacitance was increased up to 12.7 F/g, which is attributed to the porous structure of the material that offer more surface area to accumulate charge on the interfaces between the nano-filler and the matrix i.e. formation of the double layer capacitance. The nanocomposite containing 0.05% MWCNTs shows the best electrochemical performance together with excellent cycling performance, retaining 95% of its capacitance up to 5,000 cycles. Further improved biofriendly discarding ability is also proven in water medium for the nanocomposite. This novel biofriendly nanocomposite with enhanced electrochemical performance might be used as the electrode materials for environment-friendly energy storage application.

## Supporting information

**S1 Table. Optimization table.**
(DOCX)

## Author Contributions

**Conceptualization:** Muhammad Rakibul Islam.

**Data curation:** Rabeya Binta Alam, Md. Hasive Ahmad.

**Formal analysis:** Rabeya Binta Alam, Muhammad Rakibul Islam.

**Funding acquisition:** Muhammad Rakibul Islam.

**Investigation:** Rabeya Binta Alam.

**Methodology:** Rabeya Binta Alam.

**Project administration:** Muhammad Rakibul Islam.

**Resources:** Muhammad Rakibul Islam.

**Supervision:** Muhammad Rakibul Islam.

**Writing – original draft:** Rabeya Binta Alam, Muhammad Rakibul Islam.

**Writing – review & editing:** Muhammad Rakibul Islam.

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
