## [Decision Letter · Decision Letter 0]

4 Apr 2023

PONE-D-23-03580Multi-walled carbon nanotube reinforced Gelatin biopolymer with improved electrochemical performance and cyclic stability for transient energy storage applicationsPLOS ONE

Dear Dr. Islam,

Thank you for submitting your manuscript to PLOS ONE. After careful consideration, we feel that it has merit but does not fully meet PLOS ONE’s publication criteria as it currently stands. Therefore, we invite you to submit a revised version of the manuscript that addresses the points raised during the review process.

We look forward to receiving your revised manuscript.

Kind regards,

Dr. Sandeep Arya

Academic Editor

PLOS ONE

Journal Requirements:

"One of the authors, MRI gratefully acknowledges the financial support from the Ministry of science and technology, Government of Bangladesh under the grant: 39.00.0000.009.014.019-21-745. RBA and MRI are grateful to the Committee for Advanced Studies and Research (CASR), Bangladesh University of Engineering and Technology for providing financial support under the grant DAERS/R-01/CASR-337th/2021."

"One of the authors, MRI gratefully acknowledges the financial support from the Ministry of science and technology, Government of Bangladesh under the grant: 39.00.0000.009.014.019-21-745. RBA and MRI are grateful to the Committee for Advanced Studies and Research (CASR), Bangladesh University of Engineering and Technology for providing financial support under the grant DAERS/R-01/CASR-337th/2021.

Additional Editor Comments:

The authors should carefully answer all the queries mentioned by the reviewers. All the suggestions should be included.

Reviewers' comments:

Reviewer's Responses to Questions

**Comments to the Author**

1. Is the manuscript technically sound, and do the data support the conclusions?

Reviewer #1: Yes

Reviewer #2: Yes

2. Has the statistical analysis been performed appropriately and rigorously? 

Reviewer #1: Yes

Reviewer #2: No

3. Have the authors made all data underlying the findings in their manuscript fully available?

Reviewer #1: Yes

Reviewer #2: Yes

4. Is the manuscript presented in an intelligible fashion and written in standard English?

Reviewer #1: Yes

Reviewer #2: Yes

5. Review Comments to the Author

Reviewer #1: dear Authors

After carefully reading this manuscript, I would like to recommend for publishing after miner revision:

1- The English need to rechecking to avoid some mistake in Abstract and introduction

2- Update the same references in the Introduction.

Reviewer #2: 1. Title of the manuscript should be modified. It should be precise as well as catchy.

2. The optimization study for the experimental section should be added as supplementary information.

3. The steps for electrode preparation must be included in the manuscript. Also, check the working electrode that authors are using in the electrode set up. Also, check the mass of the electrode before and after the deposition of the material.

3. In figure 1, why is dip for the peak in the wavenumber 2000-3000 cm-1 (corresponding to Gel/0.01% MWCNT) not prominent?

4 The trends for the change in the morphology from Fig 2 (d-f) need explanation.

5 Mention the frequency range and voltage employed for EIS data.

6. Authors have already published one manuscript with doi.org/10.1016/j.heliyon.2021.e07468. The results and scientific explanation of the present manuscript almost resemble with the published manuscript. I did not find any novelty in the present manuscript. What is the significance of MWCNT in comparison to SWCNT? Authors should highlight all important aspect of the present work. Moreover, the analysis of the obtained experimental results need modification.

The authors should include few recent references like; Thermoelectric-powered supercapacitor based on Ni-Mn nanowires driven by quadripartite electrolyte; Role of Electrochemical techniques for photovoltaic and supercapacitor applications; etc.

6. PLOS authors have the option to publish the peer review history of their article (what does this mean?). If published, this will include your full peer review and any attached files.

Reviewer #1: **Yes: **Duha S Ahmed

Reviewer #2: No

---

## [Author Response · Author response to Decision Letter 0]

2 May 2023

Please see the attached file for the reviewer's response and list of changes.

---

## [Decision Letter · Decision Letter 1]

25 May 2023

PONE-D-23-03580R1Improved electrochemical performance of Multi-walled carbon nanotube reinforced Gelatin biopolymer for transient energy storage applicationsPLOS ONE

Dear Dr. Islam,

Thank you for submitting your manuscript to PLOS ONE. After careful consideration, we feel that it has merit but does not fully meet PLOS ONE’s publication criteria as it currently stands. Therefore, we invite you to submit a revised version of the manuscript that addresses the points raised during the review process.

We look forward to receiving your revised manuscript.

Kind regards,

Sandeep Arya

Academic Editor

PLOS ONE

Journal Requirements:

Additional Editor Comments:

Dear Authors

Please modify the manuscript as per the suggestion provided by reviewer 2. So, minor revision is necessary before final approval of the manuscript.

Reviewers' comments:

Reviewer's Responses to Questions

**Comments to the Author**

1. If the authors have adequately addressed your comments raised in a previous round of review and you feel that this manuscript is now acceptable for publication, you may indicate that here to bypass the “Comments to the Author” section, enter your conflict of interest statement in the “Confidential to Editor” section, and submit your "Accept" recommendation.

Reviewer #2: (No Response)

2. Is the manuscript technically sound, and do the data support the conclusions?

Reviewer #2: Yes

3. Has the statistical analysis been performed appropriately and rigorously? 

Reviewer #2: Yes

4. Have the authors made all data underlying the findings in their manuscript fully available?

Reviewer #2: Yes

5. Is the manuscript presented in an intelligible fashion and written in standard English?

Reviewer #2: Yes

6. Review Comments to the Author

Reviewer #2: (No Response)

7. PLOS authors have the option to publish the peer review history of their article (what does this mean?). If published, this will include your full peer review and any attached files.

Reviewer #2: No

---

## [Editor Report · Decision Letter 2]

20 Jun 2023

Improved electrochemical performance of Multi-walled carbon nanotube reinforced Gelatin biopolymer for transient energy storage applications

PONE-D-23-03580R2

Dear Dr. Islam,

We’re pleased to inform you that your manuscript has been judged scientifically suitable for publication and will be formally accepted for publication once it meets all outstanding technical requirements.

Kind regards,

Sandeep Arya

Academic Editor

PLOS ONE

Additional Editor Comments (optional):

The authors should consider the suggestions provided by reviewers with the proof read. However, the paper is provisionally accepted.
---

## [Editor Report · Acceptance letter]

6 Jul 2023

PONE-D-23-03580R2 

Improved electrochemical performance of Multi-walled carbon nanotube reinforced Gelatin biopolymer for transient energy storage applications 

Dear Dr. Islam:

I'm pleased to inform you that your manuscript has been deemed suitable for publication in PLOS ONE. Congratulations! Your manuscript is now with our production department. 

Kind regards, 

on behalf of

Dr. Sandeep Arya 

Academic Editor

PLOS ONE